# DMAF-NET: Deep Multi-Scale Attention Fusion Network for Hyperspectral Image Classification with Limited Samples

**DOI:** 10.3390/s24103153

**Published:** 2024-05-15

**Authors:** Hufeng Guo, Wenyi Liu

**Affiliations:** 1State Key Laboratory of Dynamic Measurement Technology, School of Instrument and Electronics, North University of China, Taiyuan 030051, China; b20220633@st.nuc.edu.cn; 2Department of Transportation Information Engineering, Henan College of Transportation, Zhengzhou 451460, China

**Keywords:** convolutional neural network (CNN), hyperspectral image (HSI) classification, limited samples, multi-scale feature extraction, multi-scale spatial–spectral attention, pyramidal multi-scale channel attention, multi-attention feature fusion

## Abstract

In recent years, deep learning methods have achieved remarkable success in hyperspectral image classification (HSIC), and the utilization of convolutional neural networks (CNNs) has proven to be highly effective. However, there are still several critical issues that need to be addressed in the HSIC task, such as the lack of labeled training samples, which constrains the classification accuracy and generalization ability of CNNs. To address this problem, a deep multi-scale attention fusion network (DMAF-NET) is proposed in this paper. This network is based on multi-scale features and fully exploits the deep features of samples from multiple levels and different perspectives with an aim to enhance HSIC results using limited samples. The innovation of this article is mainly reflected in three aspects: Firstly, a novel baseline network for multi-scale feature extraction is designed with a pyramid structure and densely connected 3D octave convolutional network enabling the extraction of deep-level information from features at different granularities. Secondly, a multi-scale spatial–spectral attention module and a pyramidal multi-scale channel attention module are designed, respectively. This allows modeling of the comprehensive dependencies of coordinates and directions, local and global, in four dimensions. Finally, a multi-attention fusion module is designed to effectively combine feature mappings extracted from multiple branches. Extensive experiments on four popular datasets demonstrate that the proposed method can achieve high classification accuracy even with fewer labeled samples.

## 1. Introduction

Hyperspectral images (HSIs), at the forefront of current remote sensing image technology, utilize multiple narrowband electromagnetic waves to acquire rich spatial, radiometric, and spectral information about objects of interest. With its rich information content, HSIs can be used in many fields and play a crucial role, such as: precision agriculture [1,2,3]; mineral exploration [4]; environmental detection [5,6,7]; biomedical imaging [8,9]; food safety [10,11]; urban planning [12]; military investigation [13]; climate change studies [14,15]; and many other fields. In these applications and studies, hyperspectral image classification (HSIC) plays a crucial role and has emerged as a prominent research area in the field of remote sensing and earth observation.

The task of HSIC involves assigning an appropriate class label to each pixel, thereby generating a classified map that accurately represents the distribution of land features. The conventional approach typically consists of two primary steps: feature engineering and classifier design. The first step involves manual extraction of feature information based on prior knowledge [16,17,18], followed by its classification using a classifier [19,20,21,22]. However, most traditional algorithms heavily rely on data preprocessing and manual feature extraction, which not only depend heavily on prior knowledge but also have limited generalization ability. Moreover, they solely utilize spectral information while disregarding the spatial correlation between pixels, thereby making it difficult to extract representative and discriminative features.

With the progressive advancement of remote sensing imaging technology, high-performance computing units, and computer vision theory, deep learning techniques have been employed for HSIC. This has led to a continuous enhancement in their classification accuracy [23,24]. The deep learning approaches, in contrast to conventional methods, possess the capability of automatically extracting deep abstract features from input data that are advantageous for classification tasks, thereby attaining enhanced accuracy in both classification and recognition. The stacked autoencoder (SAE) [25] and the deep belief network (DBN) [26] were initially introduced for HSIC in the field. However, these methods not only have a large number of parameters, but also require a 1D input form and suffer from loss of spatial information. To address this problem, several 2D convolutional neural network (CNN) [27,28,29] methods have been proposed, which can directly handle the 3D cubes patch of HSI. In order to further explore the spatial–spectral information in 3D HSI patches, researchers proposed a 3D CNN [30,31,32,33]. However, while 2D CNNs fail to effectively exploit the spectral dimension of HSI, 3D CNNs often encounter challenges such as a substantial parameter count, high computational complexity, and vulnerability to overfitting. Subsequently, researchers proposed a hybrid network combining both 2D CNNs and 3D CNNs [34,35,36]. This integration aims to improve the accuracy of predictive classification by leveraging advantages from both types.

With the increasing depth of networks, residual networks and densely connected networks have been successively proposed. Zhong et al. [37] proposed a spectral–spatial residual network (SSRN) based on a 3D CNN, which employs residual blocks to mitigate the issue of diminishing classification accuracy with increasing model depth, thereby facilitating gradient backpropagation. Zhang et al. [38] designed a deep residual module (DIR) for spectral–spatial feature extraction, which avoids degradation of the network while locking in the effective features at each layer. Zahisham et al. [39] proposed a two-stream residual separable convolution (2SRS) network, which utilizes deep separable convolutions to integrate residual blocks into two distinct streams for spatial and spectral processing. Dong et al. [40] proposed a two-branch cross-feedback dense network with context-aware guided attention (CFDcagaNet), which incorporates the DenseNet in a feed-forward manner to promote feature reuse and achieve higher reconstruction accuracy for super-resolution. However, pure ResNet and DenseNet suffer from a large number of parameters, high computational cost, and relatively small receptive fields per layer. Wang et al. [41] proposed a multi-scale dense connection attention network (MSDAN), which introduces multi-scale feature extraction to obtain features of different granularities using receptive fields of varying sizes. By combining DenseNet and attention mechanism, the classification performance is significantly improved. Wang et al. [42] proposed a unified multi-scale learning (UML) framework based on fully convolutional networks. In the UML framework, they introduced a multi-scale spatial channel attention mechanism and multi-scale scrubbing blocks to improve the distortion problem in land cover maps. Moreover, in our previous work [43,44], we delved into the extraction of multi-scale features and proposed two networks that incorporate attention mechanisms: the Multi-Scale Residual Network (MRA-NET) [43] and the Multi-Scale Feature Fusion Network with 3D Self-Attention (3DSA-MFN) [44]. Zhang et al. [45] introduced a classification method based on a Multi-Scale Dense Network (MSDN) with a 3D Gabor filter. Zhao et al. [46] proposed a bi-branch global+ multi-scale hybrid network (GMHN). These networks extract target features at various scales, effectively improving accuracy in HSIC tasks and demonstrating the significant value of multi-scale features in this domain.

In recent years, the Transformer model has gained widespread adoption in various intelligent large-scale applications due to its incorporation of a self-attention mechanism. This mechanism enables the model to effectively capture long-term dependencies in sequential data while demonstrating robust parallel capabilities. Sun et al. [47] proposed a Spectral–Spatial Feature Tag Transformer (SSFTT) method for HSIC, which captures spectral–spatial features and high-level semantic features, outperforming several state-of-the-art methods. Yang et al. [48] proposed an HSI Transformer (HiT) classification network that embeds convolution operations into the transformer structure to capture subtle spectral differences and convey local spatial context information. Cao et al. [49] proposed a Transformer-based MAE using contrastive learning (TMAC), which aims to combine these two methods and further improve performance. Guo et al. [50] proposed a self-supervised learning algorithm based on a spectral transformer and masking mechanism for HSIC in the presence of limited labeled data. Nevertheless, these approaches necessitate a substantial quantity of annotated samples.

The utilization of deep learning methods based on the CNN in the aforementioned exploration has significantly advanced HSIC and enhanced classification accuracy. However, these methods often necessitate a substantial number of labeled training samples to ensure effective classification, which incurs significant human resources and time costs in labeling HSI samples. Currently, the limited size of training samples often results in issues such as overfitting and reduced classification accuracy in models. The primary objective of our research is to develop a deep learning model that can effectively learn and accurately classify even with a scarcity of samples. By incorporating a deep multi-scale fusion attention mechanism, we aim to enhance the capacity for capturing subtle features in HSI, thereby improving the precision and stability of classification. The main research tasks are as follows: Firstly, design a deep learning architecture model based on CNN. This model primarily comprises modules for feature extraction, attention enhancement, information fusion, and classification. Secondly, investigate strategies for feature representation learning in scenarios with limited samples to enable the model to focus on key features and mitigate the issue of small sample sizes. Finally, evaluate and optimize the model by conducting experiments on multiple datasets to assess its performance and continuously adjust and optimize the parameters.

Inspired by the aforementioned approaches, we propose a deep multi-scale attention fusion network (DMAF-NET) based on limited training samples, aiming to fully explore the deeper and richer semantics of limited samples and improve HSIC accuracy. Firstly, the HSI data are input to the multi-scale feature backbone network to learn multilevel high-level semantics at varying granularities after data preprocessing. Subsequently, the acquired features are sequentially fed into the multi-scale spatial–spectral attention module and the multi-scale channel attention module, and the long-distance dependence of the feature map is further modeled in depth through multiple scales. Then, the multi-attention fusion module is utilized for effective feature fusion of different levels of high-level semantics. Finally, the flattened feature map is successively passed through several fully connected layers, to finally output the classification result. The main contributions of this work are as follows:A novel baseline network for multi-scale feature extraction is designed. The baseline comprises three branches. Firstly, a pyramid-like structure is employed for preliminary feature extraction to capture features at different scales. Subsequently, a dense-connected 3D octave convolutional network is utilized to learn deeper and finer-grained features within various scale windows. This allows for effective leveraging of semantic information at various levels with limited samples to extract more robust and highly generalizable features.Considering the high-resolution and multi-dimensional characteristics of HSIs, we have designed a 3D multi-scale spatial–spectral attention module and a 4D pyramid-type multi-scale channel attention module, respectively. This models the comprehensive dependencies of coordinates and directions, local and global, in four dimensions, making the model more focused on extracting information useful for classification.A multi-attention feature fusion module is designed. By fully utilizing the strong complementary and correlated information from different hierarchical features, this approach effectively integrates feature information from various levels and scales, thereby improving the performance of HSIC results under limited sample conditions.Extensive experiments based on limited labeled samples were conducted on four typical HSI datasets. The results demonstrate that the proposed DMAF-NET model outperforms other state-of-the-art deep learning-based methods in terms of both efficacy and efficiency.

The remainder of this paper is organized as follows. Section 2 describes the proposed network architecture in detail. Section 3 conducts comprehensive experiments. The ablation experiments and other impact experiments are shown in Section 4. Finally, the conclusion is drawn in Section 5.

## 2. Proposed Method

In this section, we initially present a concise introduction to the proposed DMAF-NET, followed by an elaborate exposition of each individual unit encompassed within the network.

### 2.1. Overview of the Proposed Model

The overall architecture of the DMAF-NET model proposed in this paper is illustrated in Figure 1, taking the University of Pavia dataset as a representative example. The DMAF-NET model is primarily composed of a multi-scale feature extraction backbone network, attention mechanism units, and a multi-attention feature fusion module. Considering the fact that the feature map extracted consists of four dimensions: two spatial dimensions, a spectral dimension, and a channel dimension, we have devised distinct modules to augment attention in both the space–spectral domain and channel domain; they are a 3D multi-scale spatial–spectral attention enhancement module and a 4D pyramidal multi-scale channel attention module. The DMAF-NET is an end-to-end HSIC network, in which the input is the raw HSI data $X\in R^{H\times W\times L}$, where H×W represents the spatial dimension and L represents the spectral dimension. The output is the probability of each pixel’s class in the HSI, denoted as $y\in R^{1\times 1\times c}$, where *c* represents the number of land cover classes.

Firstly, principal component analysis (PCA) [51] is performed on the raw HSI data, which can effectively reduce the dimensionality of highly redundant information and filters out bands that contribute less to classification tasks. After that, to effectively utilize the spatial and spectral information features inherent in HSI data, we extract a 3D cube consisting of neighboring pixels within a specific window size centered around the target pixel as a sample. Subsequently, the 3D cube samples are fed into a multi-scale feature extraction backbone network for deep-level learning of different granularities of features. The extracted multi-scale features are further enhanced through spatial–spectral attention and channel attention. Then, efficient fusion is performed on the three-channel multi-scale attention features. Finally, the model is classified and predicted through the fully connected layer and Softmax layer.

To optimize the DMAF-NET, we use cross-entropy as the loss function for the HSIC task, which is defined as follows:(1)$$\mathrm{Cls}=-\sum_{i=1}^{N}y_{i}\log\left(p_{i}\right)$$
where $y_{i}$ is the true class label and $p_{i}$ is the class probability predicted by the model.

### 2.2. Multi-Scale Feature Extraction Backbone Network

The term ‘multi-scale’ refers to the process of sampling signals at various levels of granularity, enabling the extraction of diverse features for accomplishing a range of tasks. In recent years, numerous studies have demonstrated the substantial advantages of multi-scale feature learning over single-scale feature learning in the domain of computer vision [52,53,54,55,56,57]. Chen et al. [58] argue that high- and low-frequency signals not only exist in natural images but also in the feature maps and channels of convolutional layers. To reduce spatial redundancy, they proposed employing lower-dimensional tensors to store slowly varying low-frequency information, and thereby introduced octave convolution. Motivated by the aforementioned content, and based on our previous work [43,44], a novel multi-scale feature extraction backbone network (MsFEBN) was proposed, as illustrated in Figure 2.

MsFEBN comprises three branches, each concurrently extracting features at distinct scales from the input feature map on its respective branch architecture, thereby enabling multi-scale feature learning. Firstly, the branch employs a pyramid network consisting of 3D convolutions with varying kernel sizes to extract multi-scale features from the input feature maps. Specifically, the kernel sizes used are 1 × 1 × 1, 3 × 3 × 3, and 5 × 5 × 5. Subsequently, the initially extracted feature maps are fed into a dense connection network based on 3D octave convolution to further facilitate deep-level feature learning across different granularities through distinct scale windows. Incorporating dense connections in a network can effectively alleviate the potential issue of gradient vanishing as the network deepens, enhancing feature propagation and reuse, ultimately leading to more robust extracted features.
(2)$$\left\{ \begin{array}{ll} X_{H} & =3\mathrm{DConv}\left(Y_{0}\right)\\ X_{L} & =3\mathrm{DConv}\left(\mathrm{DownSample}\left(Y_{0}\right)\right)\\ Y_{H} & =F_{H\to H}+F_{L\to H}\\ & =\sum {\left(W_{H\to H}\right)}^{T}X_{H}+\mathrm{Upsample}\left(\sum (W_{L\to H}{)}^{T}X_{L}\right)\\ Y_{L} & =F_{H\to L}+F_{L\to L}\\ & =\sum {\left(W_{H\to L}\right)}^{T}\mathrm{DownSample}\left(X_{H}\right)+\sum (W_{L\to L}{)}^{T}X_{L}\\ Y_{Oct1} & =𝜕\left(\beta \left(\mathrm{DownSample}\left(Y_{H}\right)\right)\right)+𝜕\left(\beta \left(Y_{L}\right)\right) \end{array}\right.$$
where $X_{H}$ and $X_{L}$ denote the factorization of feature map $Y_{0}$ into high-frequency and low-frequency components using a coefficient α; $Y_{H}$ and $Y_{L}$ denote the high-frequency and low-frequency components outputted by octave convolution; $F_{H\to H}$, $F_{L\to L}$, $F_{H\to L}$, and $F_{L\to H}$ denote intercommunication within and between frequencies, respectively; W represents the weight parameters of octave convolution; and 3DConv(⋅), β(⋅), 𝜕(⋅), DownSample(⋅), and UpSample(⋅) denote the 3Dconvolution, 3DBatchNorm, ReLu, 3DAvgPool, and Upsample functions, respectively. Subsequently, the first-level output is concatenated with the input along channels to obtain feature map $Y_{1}$ as the input for second-level 3D octave convolution. Finally, on this branch, feature map $Y_{out}$ is obtained as the ultimate result, as shown in Equations (3) and (4).
(3)$$Y_{1}=\mathrm{Cat}\left\{Y_{Oct1},\mathrm{DownSample}(Y_{0})\right\}$$
(4)$$Y_{out}=\mathrm{Cat}\{Y_{Oct2},\mathrm{DownSample}\left(Y_{0}\right),\mathrm{DownSample}(Y_{1})\}$$
where $Y_{Oct2}$ is the feature map output after the second octave convolution layer, and Cat(⋅) denotes the Concatenate function.

### 2.3. Attention Mechanism Unit

#### 2.3.1. Three-Dimensional Multi-Scale Space–Spectral Attention Enhancement Module

The internal structure of the 3D multi-scale spatial–spectral attention enhancement module (3D-MsSSAEM) is shown in Figure 3. The 3D-MsSSAEM exhibits two primary characteristics: Firstly, to optimize computational resources and expedite the learning process, channel grouping is implemented. This entails restructuring certain channels into batch dimensions, thereby dividing the channel dimension into multiple feature groups and ensuring equitable distribution of spatial semantic features within each group. Secondly, the 3D-MsSSAEM performs learning and aggregation of multi-scale spatial–spectral structural information through two branches. These two branches utilize convolutional kernels of 1 × 1 × 1 and 3 × 3 × 3, respectively, and cross-domain joint learning is performed between these two branches. This effectively establishes short-term and long-term dependencies, resulting in a stronger spatial–spectral feature extraction capability.

Specifically, after transforming certain channel dimensions into batch dimensions, the resulting sub-feature groups can be denoted as $X\in R^{H\times W\times L\times M}$, where M=C/m is the number of channels after grouping, m is the grouping factor, and the value of m is set to 20 in the proposed model. We refer to the branches with convolution kernels of 1 × 1 × 1 and 3 × 3 × 3 as the 1 × 1 × 1 branch and 3 × 3 × 3 branch, respectively. In the 1 × 1 × 1 branch, the input X is first adaptively pooled along the H-axis and W-axis in the spatial dimension and the L-axis in the spectral dimension, respectively. Subsequently, the pooled outputs are concatenated and convolved to produce ${X^{\prime }}_{1}$, as shown in Equations (5) and (6).
(5)$$\left\{ \begin{array}{ll} {\mathrm{Agp}}_{H}\left(X\right) & =\frac{1}{W\times L}\sum_{i,j}^{W,L}x_{ij}\\ {\mathrm{Agp}}_{W}\left(X\right) & =\frac{1}{H\times L}\sum_{i,j}^{H,L}x_{ij}\\ {\mathrm{Agp}}_{L}\left(X\right) & =\frac{1}{H\times W}\sum_{i,j}^{H,W}x_{ij} \end{array}\right.$$
(6)$$X_{1}^{\prime }=3\mathrm{DConv}\left(\mathrm{Cat}\left\{P\left({\mathrm{Agp}}_{H}\left(X\right)\right),P\left({\mathrm{Agp}}_{W}\left(X\right)\right),{\mathrm{Agp}}_{L}\left(X\right)\right\}\right)$$
where Agp(⋅) and P(⋅) denote the AdaptiveAvgPool and Permute function, respectively. After that, decompose $X_{1}^{\prime }$ back into three vectors, input them separately into the Sigmoid function, multiply to obtain the first attention map, and reassign weights to the original input X, resulting in output $X_{1}^{\prime \prime }$, as shown in Equation (7).
(7)$$\left\{ \begin{array}{l} X_{H}^{\prime },X_{W}^{\prime },X_{L}^{\prime }=\mathrm{Split}\left(X_{1}^{\prime }\right)\\ X_{1}^{\prime \prime }=X\times \mathrm{Sig}\left(X_{H}^{\prime }\right)\times \mathrm{Sig}\left(X_{W}^{\prime }\right)\times \mathrm{Sig}\left(X_{L}^{\prime }\right) \end{array}\right.$$
where Sig(⋅) denotes the Sigmoid function. Then, cross-domain collaborative learning is performed between the 1 × 1 × 1 branch and 3 × 3 × 3 branch to model the dependencies between diverse scales, local and global. The second and third attention maps, denoted as $X_{1}^{\prime \prime \prime }$ and $X_{2}^{\prime \prime }$, respectively, are generated based on this process, as shown in Equation (8).
(8){X1‴=X2′×Soft(Re(Agp(Gn(X1″))))X2″=Gn(X1′)×Soft(P(Re(Agp(X2′))))
where Gn(⋅), Re(⋅), and Soft(⋅) denote the GroupNorm, Reshape, and Softmax function. Finally, the attention maps of the two branches are summed element-wise and the fourth attention map is obtained by the Sigmoid function, and then the original inputs are reweighted, as shown in Equation (9).
(9)$$X_{out}=X\times \mathrm{Sig}\left(X_{1}^{\prime \prime \prime }+X_{2}^{\prime \prime }\right)$$

#### 2.3.2. Four-Dimensional Pyramid-Style Multi-Scale Channel Attention Module

SENet [59] is a highly representative channel attention architecture method, which incorporates the SE module. It models the internal feature maps in the channel dimension through global average pooling and two fully connected layers with non-linear activations, effectively capturing the interdependencies among channels of the feature maps. However, the SE module only simply employs global average pooling to map spatial feature information to a low-dimensional space, overlooking the intricate structural details within the feature map. In light of this, we have optimized the SE module by incorporating a multi-scale pyramid encoding structure and proposed a 4D pyramid-style multi-scale channel attention module (4D-PMsCAM) suitable for HSIC, as shown in Figure 4. The 4D-PMsCAM employs a three-layer pyramid structure for encoding, which allows the integration of spatial information at different scales, thereby constructing richer structural information and establishing longer-distance channel dependencies.

The encoding structure within the module is composed of three 3D adaptive average pooling layers with varying scales, resembling a pyramid shape. Specifically, the output window sizes are 1 × 1 × 1, 2 × 2 × 2, and 4 × 4 × 4, respectively. The three outputs are subsequently reshaped and concatenated along the channel dimension to obtain a vector $z\in R^{1\times 1\times 1\times C}$. The subsequent steps are shown in Equation (10).
(10)$$s=F\left(z,W\right)=\mathrm{Sig}\left(W_{2}𝜕\left(W_{1}z\right)\right)$$
where $W_{1}\in R^{\frac{c}{r}\times C}$ and $W_{2}\in R^{\frac{c}{r}\times C}$ denote the weight parameters of the two fully connected layers, respectively, r is the channel dimension reduction factor, and s represents the final dependency relationship of the channel dimensions obtained.

### 2.4. Multi-Attention Feature Fusion Module

SKNets [60] is a learning network that effectively captures objects with varying scales and incorporates a module known as Selective Kernel Convolution (SK convolution). The SK module comprises two branches with varying sizes of convolutional kernels, employing a non-linear approach to effectively aggregate information across multiple scales. This adaptive mechanism enables neurons to dynamically adjust the receptive field size. Motivated by the concept of SK convolution, we propose a novel multi-attention feature fusion module (MAFFM) for HSIC, as illustrated in Figure 5.

Before inputting the MAFFM, the three feature maps are concatenated into one feature map along the channel dimension. Therefore, inside the MAFFM, the input feature map needs to be split first to restore the original feature map of the three branches. Subsequently, the three features are recombined through a straightforward element-wise summation for aggregation. Then, the integration of spatial information is further enhanced by learning to weight features and regulating the flow of information through gating mechanisms. Finally, the feature fusion of each branch is achieved based on the weights of each branch. The average pooling method is commonly employed for spatial information aggregation. However, by incorporating the max pooling method, an additional crucial clue regarding distinct object features can be gathered [61], thereby enabling the inference of more refined attention information in the feature map. Specifically, assuming the three branches of splitting and restoring are A, B, and C $\in R^{H\times W\times L\times C}$, firstly, the three branches are aggregated to obtain a feature map U. Concurrently, spatial compression is applied to the feature map U through 3D global average pooling and 3D global maximum pooling, resulting in a feature vector $z\in R^{1\times 1\times 1\times C}$, and the compressed feature vector $z^{\prime }\in R^{1\times 1\times 1\times \frac{C}{r}}$ is obtained after passing through a fully connected layer. The process is depicted by Equations (11)–(13).
(11)U=A+B+C
(12)z=Agp(U)+Mxp(U)
(13)$$z^{\prime }=F\left(z,W\right)=𝜕\left(\beta \left(Wz\right)\right)$$
where Mxp(⋅) denotes the 3DAdaptiveMaxPool function, and $W\in R^{\frac{c}{r}\times C}$. The Softmax function is subsequently employed to compute the weights assigned to each branch. Ultimately, the fused output feature map $V\in R^{H\times W\times L\times C}$ is obtained by reconfiguring and summing up each branch based on its corresponding weigh, as shown in Equations (14) and (15).
(14)$$\left\{ \begin{array}{l} a_{c}=\frac{e^{{A^{\prime }}_{c}z^{\prime }}}{e^{{A^{\prime }}_{c}z^{\prime }}+e^{{B^{\prime }}_{c}z^{\prime }}+e^{{C^{\prime }}_{c}z^{\prime }}}\\ b_{c}=\frac{e^{{B^{\prime }}_{c}z^{\prime }}}{e^{{A^{\prime }}_{c}z^{\prime }}+e^{{B^{\prime }}_{c}z^{\prime }}+e^{{C^{\prime }}_{c}z^{\prime }}}\\ c_{c}=\frac{e^{{C^{\prime }}_{c}z^{\prime }}}{e^{{A^{\prime }}_{c}z^{\prime }}+e^{{B^{\prime }}_{c}z^{\prime }}+e^{{C^{\prime }}_{c}z^{\prime }}} \end{array}\right.$$
(15)$$V_{c}=a_{c}A_{c}+b_{c}B_{c}+c_{c}C_{c},a_{c}+b_{c}+c_{c}=1$$
where $A^{\prime }$, $B^{\prime }$, and $C^{\prime }\in R^{C\times \frac{c}{r}}$. *a*, *b*, and *c* denote the soft attention features of A, B, and C, respectively. Note that $A_{c}^{\prime }\in R^{1\times \frac{c}{r}}$ is the c-th row of $A^{\prime }$, $a_{c}$ is the *c*-th element of *a*, and others with c subscripts are similar.

## 3. Experiments and Results

In this section, we initially present the four classic datasets utilized and elucidate the experimental setup. Subsequently, a comprehensive analysis of crucial network hyperparameters is conducted. Then, we perform quantitative and qualitative experiments to compare and analyze our proposed model against other state-of-the-art methods. Finally, ablation experiments as well as other impact studies are conducted.

### 3.1. Dataset Description

The HSI dataset is acquired through remote sensing satellites or unmanned aerial vehicles, and undergoes preprocessing procedures including atmospheric correction, denoising, band selection, and feature extraction. Subsequently, it is annotated for classification purposes to generate training and testing datasets. In order to assess the efficacy and generalizability of the proposed network model across diverse HSI datasets, we selected four challenging public HSI datasets: Salinas scene (SA), University of Pavia (UP), Indian Pines (IP), and WHU-Hi-LongKou (LK). Among them, the SA, UP, and IP datasets can be obtained from: https://www.ehu.eus/ccwintco/index.php?title=Hyperspectral_Remote_Sensing_Scenes (accessed on 2 May 2024), and the LK dataset can be obtained from: http://rsidea.whu.edu.cn/resource_WHUHi_sharing.htm (accessed on 2 May 2024). The summarized details of these four datasets are presented in Table 1.

In summary, these datasets exhibit variations in terms of spatial scale, spectral range, and land cover categories. For instance, the SA and LK datasets encompass multiple spectral dimensions, diverse land cover categories, and a substantial number of samples. The UP dataset possesses larger spatial dimensions, high spatial resolution, fewer spectral dimensions, a limited number of land cover categories, and more samples. Conversely, the IP dataset presents smaller spatial dimensions with lower spatial resolution while incorporating multiple spectral dimensions alongside diverse land cover categories; nevertheless, it suffers from scarcity in sample quantity, which is further exacerbated by extreme imbalance among them. Through this selection process that encompasses four different application scenarios, our model’s performance will be validated. For visual representation purposes, the pseudo-color images and the corresponding ground truth maps for these datasets are shown in Figure 6. The SA dataset’s ground cover consists mainly of crops such as bitumen, fallow, stubble, celery, lettuce, and vineyard. The UP dataset consists mainly of buildings such as asphalt, gravel, and bitumen. The IP dataset consists mainly of vegetation such as alfalfa, corn, and soybean, etc. The LK dataset mainly includes crops such as corn, cotton, and rice. In addition, Table 2, Table 3, Table 4 and Table 5 provide detailed information on the distribution of samples for each category in each of the four datasets, as well as the number of training samples, respectively.

### 3.2. Experimental Settings

The experiments presented in this article were conducted on a computer system equipped with an NVIDIA GeForce RTX 2060 SUPER, Intel^®^ CoreTM i7-9700F @3.00GHz × 8 CPU, and 32GB of RAM. The proposed DMAF-NET was implemented using PyTorch 1.10, Python 3.8.17, Keras 2.10, and Numpy 1.23.5 within a Linux (Ubuntu 18.04.6) operating system environment.

In the training phase, we employed Adam as the optimization algorithm with a learning rate of 0.001, batch size of 128, training epoch set to 100, and dropout set to 0.4. Additionally, a comprehensive analysis on critical hyperparameters such as patch size and PCA component settings will be conducted in Section 3.3.

In the experiment, Overall Accuracy (OA), Average Accuracy (AA), and Kappa coefficient (Kappa) are utilized as quantitative metrics to evaluate the performance of each method. OA represents the ratio of correctly classified pixels to total pixels. AA denotes the average accuracy of classification for each category. Kappa is a statistical measure that assesses the alignment between model predictions and actual classification results, thereby reflecting the overall effectiveness of the classifier. In addition, the experimental results in the article were obtained by running the experiment 10 times with the same random seed and calculating the average.

### 3.3. Parametric Analysis

In practical scenarios, when a substantial number of samples are available, the sensitivity of hyperparameters to classification results tends to be relatively diminished. However, in this study, we pay more attention to the classification performance under limited sample conditions, so some hyperparameters’ tuning is particularly important. During the data preprocessing stage, the spatial patch size and the optimal number of spectral components retained after PCA processing are critical hyperparameters that significantly influence both network training and final classification performance. The optimal parameter configurations of the proposed DMAF-NET for the four datasets are shown in Table 6.

#### 3.3.1. Analysis of the Patch Size

The patch size refers to the rectangular size of the image space inputted into training and prediction modules. This rectangular image is centered on the sample pixel point, and its size determines the amount of spatial information contained around the sample pixel that is utilized for classification. In order to configure the appropriate patch size and assess its impact on the performance of our proposed network, we set the patch size = {(8 × 8), (12 × 12), (16 × 16), (20 × 20), (24 × 24), and (28 × 28)}, respectively; the impact of six sets of parameters was studied, and the results are shown in Figure 7. The experiments were conducted under fixed conditions of PCA components (SA = 28, UP = 20, IP = 32, LK = 16) and a consistent number of training samples (10 samples per class for each dataset).

In the SA dataset, as the patch size increases from (8 × 8) to (24 × 24), there is a gradual improvement observed in the three evaluation metrics of OA, AA, and Kappa. However, when the patch size is set to (28 × 28), both OA and Kappa exhibit a decrease. The evaluation metrics OA, AA, and Kappa in the UP, IP, and LK datasets exhibit a trend of initially increasing and subsequently decreasing as the patch size gradually increases, and these datasets peaked at (16 × 16), (20 × 20), and (24 × 24), respectively. Taken together, the patch sizes for the SA, UP, IP, and LK datasets were set as (24 × 24), (16 × 16), (20 × 20), and (24 × 24), respectively. The experimental results are consistent with the theory. The SA and LK datasets encompass large areas of farmland, exhibiting a well-balanced distribution of land cover categories and high concentration, which manifests in patchy patterns. Consequently, employing larger patch sizes proves advantageous for training and learning. Conversely, the UP dataset was shot on a university campus where land cover is more dispersed and less concentrated; thus, it is not advisable to employ excessively large patch sizes. In addition, it can be seen from the figure that the training time increases with the increase in the patch size.

#### 3.3.2. Analysis of the PCA Components

Since the proposed DMAF-NET uses the PCA algorithm to reduce the spectral dimension in the data preprocessing stage, it achieves a balance between removing redundant information and retaining effective features. Therefore, the number of components retained after PCA (PCA components) has a great impact on the training learning and final classification results of the model. In the experiment, we conducted an analysis and study on different PCA components, and the experimental results are shown in Figure 8. The experiments were conducted under fixed conditions of patch sizes (SA = 24, UP = 16, IP = 20, LK = 24) and a consistent number of training samples (10 samples per class for each dataset). In these four datasets, as the number of PCA components increases, the three evaluation metrics OA, AA, and Kappa generally exhibit an initial increase followed by a subsequent decrease trend, reaching their peak values at 24, 24, 44, and 16, respectively. In addition, it can be observed from the figure that the training time increases as the number of PCA components increases.

### 3.4. Comparison with Other Methods

To substantiate the superiority of the proposed method, a comparative analysis is conducted with other prominent HSIC methods proposed in recent years, namely: the 3D-CNN [30], HybridSN [34], SSRN [37], Tri-CNN [62], MCNN-CP [35], SSFTT [47], and Oct-MCNN-HS [63]. The 3D-CNN is a 3D finite element model based on a CNN that incorporates regularization techniques to extract effective spatial–spectral features from HSIs. The HybridSN employs a hybrid architecture that integrates a 3D CNN and 2D CNN to concurrently extract spatial–spectral feature information, thereby mitigating the model’s complexity compared to solely relying on the 3D CNN. The SSRN is a deep network based on a 3D CNN, it incorporates residual connections to mitigate the gradient problem encountered by deeper networks in other deep learning methods, thereby enhancing classification accuracy. The Tri-CNN is a three-branch feature fusion network based on a multi-scale 3D-CNN. SSFTT is built upon the widely adopted Transformer architecture, enabling effective extraction of spatial–spectral features and high-level semantic representations, thereby achieving remarkable classification performance. MCNN-CP integrates the covariance pooling technique with the HybridSN, facilitating the extraction of second-order information from spatial–spectral feature maps. Oct-MCNN-HS is based on the MCNN-CP architecture and incorporates a combination of a 3D octave and 2D Vanilla CNN. By utilizing synchronized shift operations to aggregate information from the same spatial positions along the channel dimension, it ensures more compact feature representation. To ensure the fairness of the comparative evaluation experiments, all comparative experiments were strictly conducted following the parameter configuration and experimental procedures stated in the original text.

#### 3.4.1. Evaluation Results with a Training Sample Limit of 10 for Each Category

The classification results of each method on the SA, UP, IP, and LK datasets are presented in Table 7, Table 8, Table 9 and Table 10, encompassing OA, AA, and Kappa, as well as the classification accuracy for each category. As can be seen from these tables, the classification accuracy of both the 3D-CNN and the HybridSN is relatively low due to their limited integration of spatial–spectral features, achieved solely through convolutions. The SSRN, MCNN-CP, and Tri-CNN can effectively integrate spatial–spectral features with fair classification accuracy. The SSFTT and Oct-MCNN-HS models are able to establish global or local dependencies, thus achieving better classification performance. Obviously, the proposed model effectively integrates feature information from multiple scales and captures long-range dependencies among spatial–spectral channels, resulting in superior classification accuracy across all datasets. Specifically, in the SA dataset, all methods achieved a classification accuracy exceeding 90%. The classification accuracies of the 3D-CNN and HybridSN even reach those of the SSRN and MCNN-CP. The proposed model achieved 97.2%, 98.3%, and 96.9% for OA, AA, and Kappa, respectively, which were higher than the second-place SSFTT by 3%, 1.3%, and 3.3%. In the UP dataset, the 3D-CNN exhibits the lowest classification accuracy, whereas Oct-MCNN-HS demonstrates superior performance compared to SSFTT. Our proposed model achieves 90.12%, 90.75%, and 87.3% for OA, AA, and Kappa, respectively, surpassing Oct-MCNN-HS by 3.32%, 1.45%, and 4.3%, correspondingly. In the IP dataset, the 3D-CNN is also the worst, and SSFTT and Oct-MCNN-HS are comparable. The OA, AA, and Kappa of our proposed model reach 81.8%, 89.2%, and 79.5%, respectively, which are 2.75%, 0.73%, and 3.07% higher than the second place, respectively. The classification accuracies of the proposed model are also the highest in the LK dataset with 96.24%, 96.81%, and 95.70% for OA, AA, and Kappa, respectively. Overall, the model proposed in this paper achieves a high performance level on each of the four datasets with different characteristics. This is due to the fact that the model is based on multi-scale feature extraction and attention enhancement, which enables it to better discriminate the spatial–spectral feature information and has good robust performance, thus greatly alleviating the overfitting problem under fewer samples and conditions.

The visual results of land cover classification using each method on the SA, UP, IP, and LK datasets are illustrated in Figure 9, Figure 10, Figure 11 and Figure 12. From these classification maps, it can be seen that the 3D-CNN and HybridSN are the least effective, with more misclassifications and the most noisy points; SSFTT and Oct-MCNN-HS have higher classification accuracies and relatively clearer feature boundaries; and our proposed method has the highest classification accuracy and the best fidelity. This is consistent with the quantitative comparison results in Table 7, Table 8, Table 9 and Table 10. Specifically, in the SA dataset, all methods misclassify to varying degrees in the vineyard and lettuce regions in the upper left of the dataset. Among them, the SRNN has more pepper noise points, while our proposed method clearly achieves the best classification performance with only a few misclassifications in the vineyard region. In the UP dataset, the 3D-CNN and HybridSN had more misclassifications in the intermediate meadow and bare soil regions of the dataset; the other methods had some misclassifications in the intermediate bare soil and bottom grass regions, and our proposed method had the least number of misclassifications. In the IP dataset, the 3D-CNN, HybridSN, SSRN, and Tri-CNN have the most misclassification errors in the corn and soybean areas in the upper left corner of the dataset, and there are large areas of category confusion. SSFTT and Oct-MCNN-HS have fewer misclassifications and the boundaries of ground objects are relatively clear. However, our proposed method clearly outperforms them all. In the LK dataset, all methods in the middle region of the dataset have more misclassifications, except our proposed method, which has the least misclassifications.

#### 3.4.2. Evaluation Results with Different Training Sample Sizes

In order to comprehensively assess the effectiveness and superiority of the classification model, we also evaluated it on the SA, UP, IP, and LK datasets based on different numbers of training samples. We conducted a study employing two sampling strategies: fixed quantity sampling and fixed proportion sampling.

The classification results obtained by fixed quantity sampling from each category are illustrated in Figure 13. For the SA, UP, and LK datasets, a random selection of 5, 10, 15, and 20 samples was made for analysis and research in each category. Considering the serious imbalance in the number of categories within the IP dataset (with only 20 samples available for oats), a random selection of 5, 10, and 15 samples was conducted for analysis and research. The graphs reveal that our proposed model exhibits exceptional performance across all three datasets, surpassing other methods in terms of OA, AA, and Kappa. Figure 14 illustrates the classification results obtained through sampling with a fixed proportion. In the SA, UP, and LK datasets, random sampling was performed at 0.1%, 0.2%, 0.5%, 1%, and 5% of the sample size for each category, respectively, whereas for the IP data, random sampling was performed at 5%, 10%, and 15% of the sample size for each category (when less than 5%, oats and grass-pasture-mowed were not sampled). Figure 14(a-1–a-3,b-1–b-3) indicate that all methods exhibit a rapid increase in classification accuracy between 0.1% and 0.5% of the sample quantity. Once the sample reaches 0.5%, the classification accuracy plateaus, resulting in slower improvements with further increases in sample quantity. In the UP dataset, the proposed method exhibits slightly inferior performance compared to the SSRN when trained with only 0.1% of the samples; however, it outperforms all other methods when beyond 0.2%.

Comparing the experimental results of the two sampling strategies, our proposed model shows more significant advantages compared to other models in the comparative experiment with a fixed quantity sampling strategy across all datasets. However, when it comes to proportional sampling, if the training sample size is less than 0.5%, due to the limited number of trainable samples provided by certain categories with low total counts in imbalanced datasets, the classification accuracy of all classification methods tends to be lower.

#### 3.4.3. Computational Complexity

The number of total parameters of the model and the training time have always been two important metrics in assessing the computational complexity of the model. The total parameters and training time of all models on the SA, UP, and IP datasets are summarized in Table 11. The 3D-CNN has the highest number of parameters, and the SSRN consumes the longest training time. SSFTT is based on the Transformer structure with parallel computing capability, so its training time is short and the number of parameters is small. Our proposed method is second only to SSFTT in terms of training time, and the number of parameters is much lower than that of the 3D-CNN, HybridSN, and Oct-MCNN-HS. The results indicate that our method exhibits the characteristics of low computational resource consumption and rapid convergence speed in limited sample classification tasks.

## 4. Discussion

In this section, we conducted ablation experiments and some other influential studies to evaluate the efficacy of the proposed model. The experiments in this section were conducted based on a statistical analysis of 10 samples for each category.

### 4.1. Ablation Studies

In order to further investigate the potential contributions of individual units within the proposed model, an independent role analysis was conducted in this subsection. The proposed DMAF-NET network comprises the following key components: the multi-scale feature extraction backbone network (MsFEBN), 3D multi-scale space–spectral attention enhancement module (3D-MsSSAEM), 4D pyramid-style multi-scale channel attention enhancement module (4D-PMsCAM), and multi-attention feature fusion module (MAFFM). In the DMAF-NET framework, we systematically eliminated each of the aforementioned components and conducted impact experimental studies. The experimental results are shown in Figure 15. These graphs demonstrate that each component contributes positively to the classification task. Firstly, when utilizing only the MsFEBN without incorporating any attention mechanism or feature fusion module, the classification accuracies on the SA, UP, and IP datasets are 95.6%, 86.6%, and 79.3%, respectively. These results demonstrate that the MsFEBN is well-suited for efficient feature extraction in limited sample classification tasks. Second, the classification accuracy decreases on all three datasets when the 3D-MsSSAEM or 4D-PMsCAM are not utilized. Additionally, the elimination of both attention mechanisms leads to a further decline in classification accuracy. Specifically, the SA, UP, and IP datasets exhibit decreases of 1.5%, 0.9%, and 2%, respectively with accuracies dropping to 95.7%, 89.2%, and 79.8%. These results demonstrate that both attention mechanisms enhance important information while suppressing redundant information at different levels and depths. Furthermore, the absence of the MAFFM resulted in reductions in classification accuracy by 0.5%, 1.5%, and 1.3% for the SA, UP, and IP datasets, respectively. This observation highlights the effective integration of semantic information from diverse levels and perspectives by the MAFFM, leading to improved classification accuracy even with limited samples.

### 4.2. Other Impact Studies

#### 4.2.1. The Influence of Different Size Convolution Kernels in Three Branches of Baseline

To substantiate the rationality of selecting diverse sizes of convolutional kernels across the three branches in our proposed MsFEBN, we conducted comparative experiments encompassing varying kernel sizes. As illustrated in the previously mentioned Figure 2, the three branches were configured with convolutional kernels of 1 × 1 × 1, 3 × 3 × 3, and 5 × 5 × 5, respectively; let us denote them as 1–3–5. In the comparative experiment, we also employed four additional combinations: 3 × 3 × 3, 5 × 5 × 5, and 7 × 7 × 7, denoted as 3–5–7; similarly, there were also 1–3–7, 1–5–7, and 3–3–3. The experimental findings are depicted in Figure 16. The figure reveals that the 1–3–5 combination has the highest classification accuracy, followed by the 3–5–7 combination, and the other three combinations have relatively low accuracy. This also confirms that choosing the appropriate scale for multi-scale learning is superior to single-scale learning. It can also be seen from the figure that the training times for the 1–3–5 combination and the 3–3–3 combination are the lowest across all three datasets. This is because the computational time consumed increases with the increase in convolutional kernel size.

#### 4.2.2. The Influence of Varying Numbers of 3D Octave Convolutions in Three Branches of Baseline

We conducted experimental verification of the number of cascaded 3D octave convolutions in the three branches of the MsFFBN. The experiment compared the effectiveness of one, two, and three 3D octave convolutions in each branch. The experimental results are presented in Figure 17. The classification accuracy in all three datasets reached its peak when employing a cascade of two 3D octave convolutions. Simultaneously, with an increase in the number of cascades, the network depth gradually expanded, resulting in a corresponding rise in computational time.

#### 4.2.3. The Influence of Different Dimensionality Reduction Method

Based on the model proposed in this article, we conducted comparative experiments on three dimensionality reduction methods, namely FastICA [64], FactorAnalysis [65], and PCA. The experimental results are presented in Figure 18. The results consistently demonstrate that FastICA exhibits the poorest performance across all three datasets, followed by FactorAnalysis, while PCA attains the highest level of accuracy. This is mainly due to the fact that FastICA has higher requirements for non-Gaussianity and non-linear independence of the data, and FactorAnalysis is also not applicable to non-linearly correlated data, as well as having a more complex process of selecting the appropriate number of factors, which makes FastICA and FactorAnalysis perform poorly on relatively linear HSI data. However, PCA is a linear dimensionality reduction technique that effectively preserves the maximum variance in HSIs, thereby retaining the essential characteristics of the data and capturing their overall structure with high efficacy.

## 5. Conclusions

This article explores the application of CNNs in deep learning to HSIC. Considering that obtaining sufficient labeled training samples for HSIC is a costly and time-consuming task, we propose a deep multi-scale attention network suitable for limited training sample conditions. The network primarily comprises a multi-scale feature extraction backbone network, an attention mechanism unit, and a feature fusion unit. The attention mechanism units consist of a 3D multi-scale spatial–spectral attention module and a 4D pyramid-style multi-scale channel attention module. In the network, the multi-scale feature extraction backbone network possesses the capability to extract features at various levels and granularities; attention units effectively model long-range dependencies in feature mappings across multiple scales, accentuating significant contributions while suppressing redundant features; and the fusion module effectively fuses high-level semantic information from multiple branches at different levels. Extensive experiments were conducted on four publicly available datasets. A comprehensive comparison was performed with seven other methods, namely the 3D-CNN, HybridSN, SSRN, Tri-CNN, SSFTT, MCNN-CP, and Oct-MCNN-HS. The results demonstrate that the proposed DMAF-NET model attains superior classification accuracy, enhanced robustness, and improved generalization capability.

The DMAF-NET model, however, is based on supervised training and does not fully mitigate the limitations imposed by annotated labels. Moreover, in comparison to the widely adopted Transformer architecture, it exhibits marginally lower efficiency. In the future, we will investigate the potential of incorporating semi-supervised or unsupervised methods in HSIC to further diminish reliance on annotated samples for this task.

## Figures and Tables

**Figure 1 sensors-24-03153-f001:**
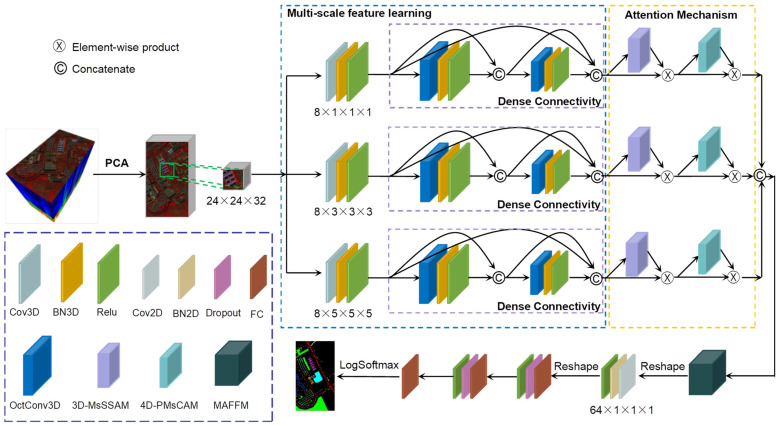
Architecture of the proposed DMAF-NET.

**Figure 2 sensors-24-03153-f002:**
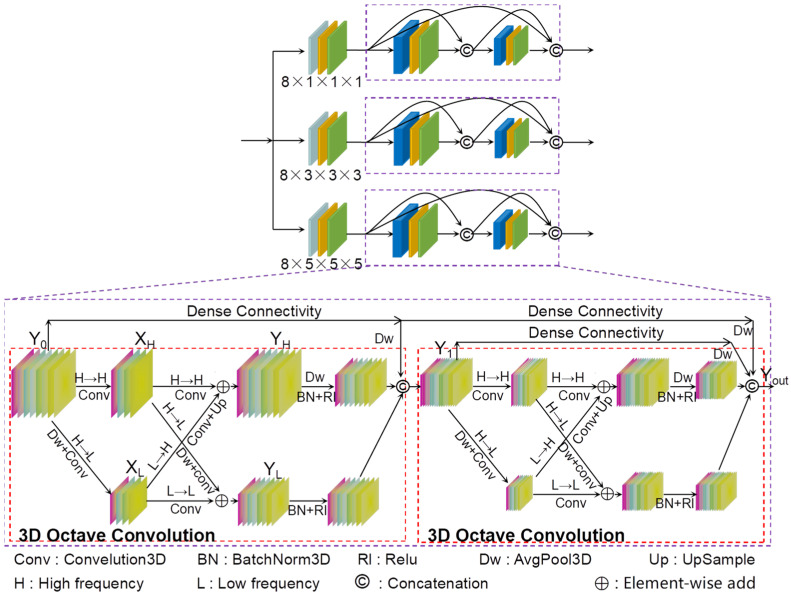
Multi-scale feature extraction backbone network.

**Figure 3 sensors-24-03153-f003:**
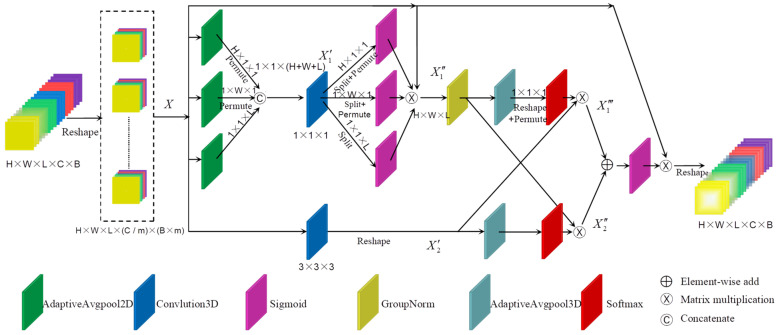
Three-dimensional multi-scale space–spectral attention enhancement module.

**Figure 4 sensors-24-03153-f004:**
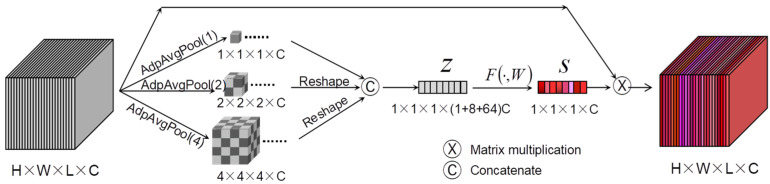
Four-dimensional pyramid-style multi-scale channel attention module.

**Figure 5 sensors-24-03153-f005:**
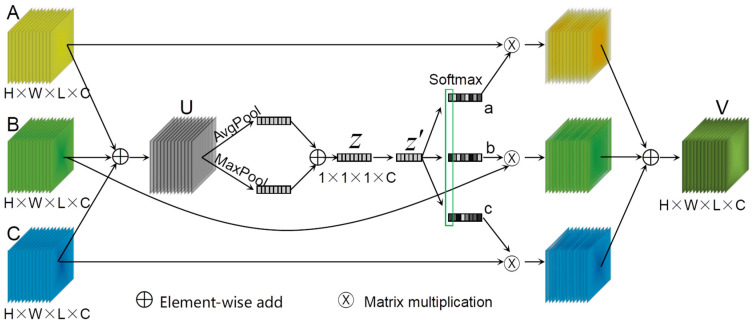
Multi-attention feature fusion module.

**Figure 6 sensors-24-03153-f006:**
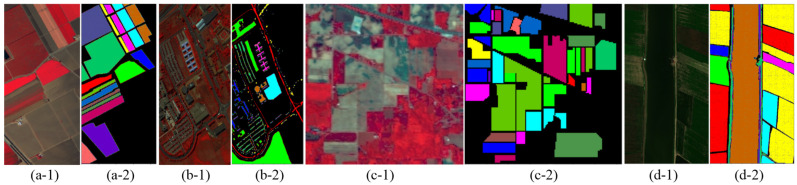
The pseudo-color images and the corresponding ground truth maps for the SA, UP, and IP datasets. (**a-1**) Pseudo-color map of SA. (**a-2**) Ground truth map of SA. (**b-1**) Pseudo-color map of UP. (**b-2**) Ground truth map of UP. (**c-1**) Pseudo-color map of IP. (**c-2**) Ground truth map of IP. (**d-1**) Pseudo-color map of LK. (**d-2**) Ground truth map of LK.

**Figure 7 sensors-24-03153-f007:**
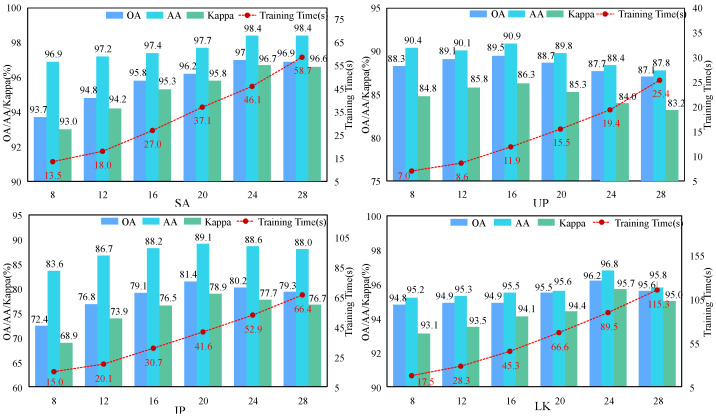
Classification results (%) and training time (seconds) for each dataset under different patch sizes.

**Figure 8 sensors-24-03153-f008:**
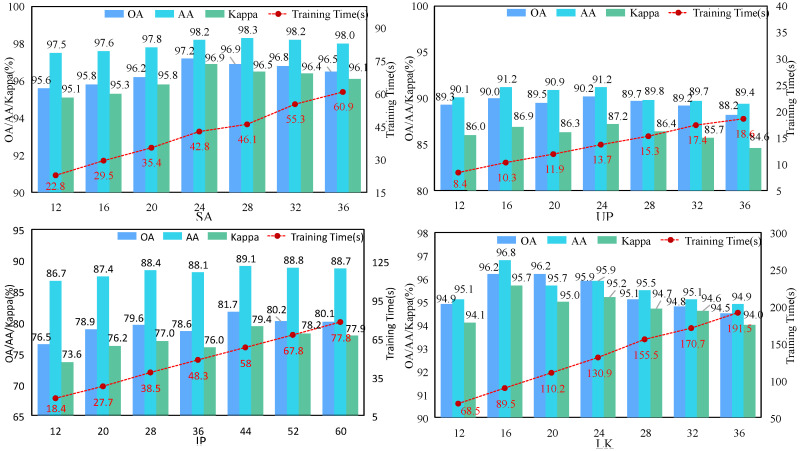
Classification results (%) and training time (seconds) for each dataset with a different number of components retained during PCA operation.

**Figure 9 sensors-24-03153-f009:**
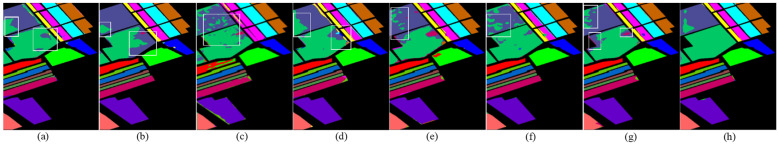
Classification maps generated by all of the competing methods on the SA dataset with 10 training samples for each category. (**a**) 3D-CNN. (**b**) HybridSN. (**c**) SSRN. (**d**) Tri-CNN. (**e**) MCNN-CP. (**f**) SSFTT. (**g**) Oct-MCNN-HS. (**h**) Proposed method.

**Figure 10 sensors-24-03153-f010:**
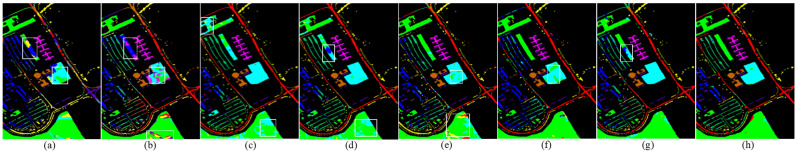
Classification maps generated by all of the competing methods on the UP dataset with 10 training samples for each category. (**a**) 3D-CNN. (**b**) HybridSN. (**c**) SSRN. (**d**) Tri-CNN. (**e**) MCNN-CP. (**f**) SSFTT. (**g**) Oct-MCNN-HS. (**h**) Proposed method.

**Figure 11 sensors-24-03153-f011:**
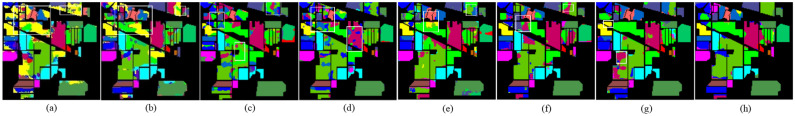
Classification maps generated by all of the competing methods on the IP dataset with 10 training samples for each category. (**a**) 3D-CNN. (**b**) HybridSN. (**c**) SSRN. (**d**) Tri-CNN. (**e**) MCNN-CP. (**f**) SSFTT. (**g**) Oct-MCNN-HS. (**h**) Proposed method.

**Figure 12 sensors-24-03153-f012:**
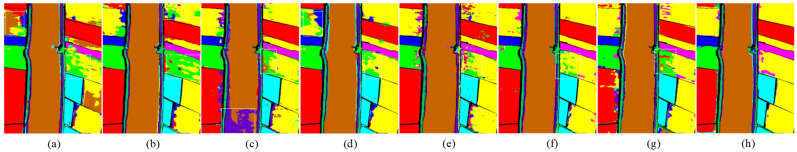
Classification maps generated by all of the competing methods on the LK dataset with 10 training samples for each category. (**a**) 3D-CNN. (**b**) HybridSN. (**c**) SSRN. (**d**) Tri-CNN. (**e**) MCNN-CP. (**f**) SSFTT. (**g**) Oct-MCNN-HS. (**h**) Proposed method.

**Figure 13 sensors-24-03153-f013:**
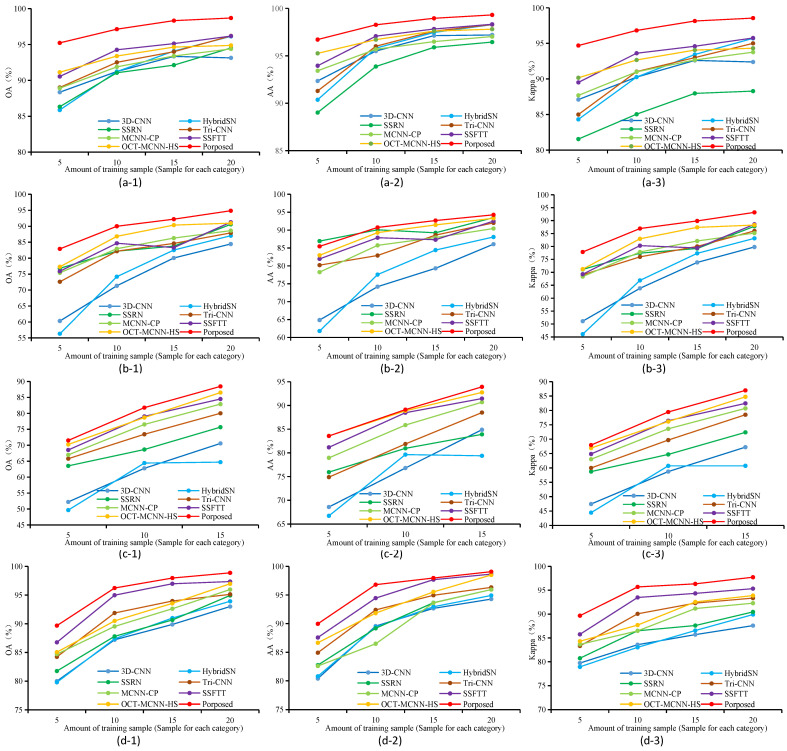
Classification results (%) for all competing methods using different amount of training samples on the three datasets; fixed quantity sampling is used for each category. (**a-1**) The OA of SA dataset. (**a-2**) The AA of SA dataset. (**a-3**) The Kappa of SA dataset. (**b-1**) The OA of UP dataset. (**b-2**) The AA of UP dataset. (**b-3**) The Kappa of UP dataset. (**c-1**) The OA of IP dataset. (**c-2**) The AA of IP dataset. (**c-3**) The Kappa of IP dataset. (**d-1**) The OA of LK dataset. (**d-2**) The AA of LK dataset. (**d-3**) The Kappa of LK dataset.

**Figure 14 sensors-24-03153-f014:**
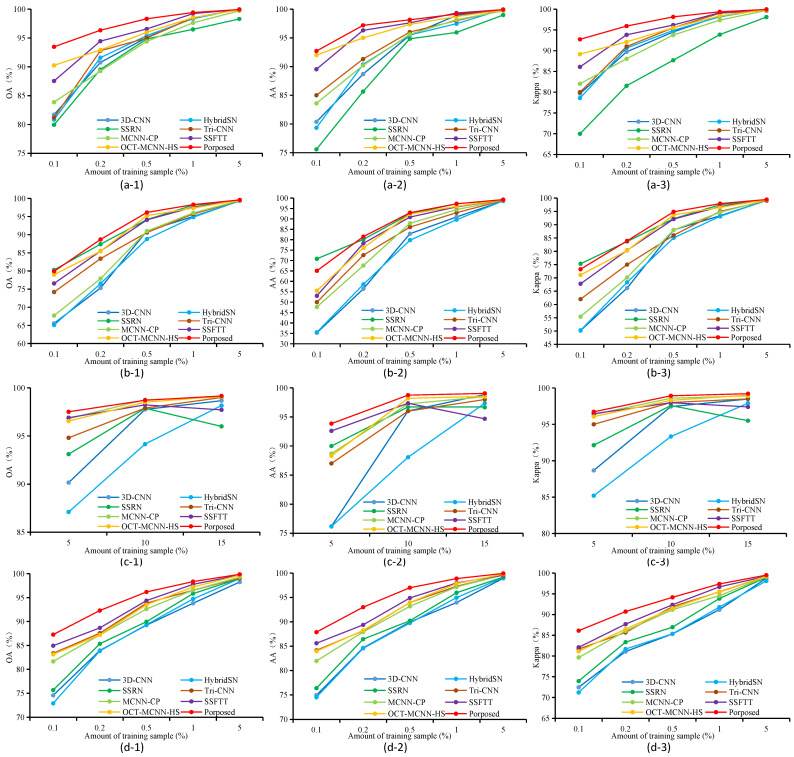
Classification results (%) for all competing methods using different amount of training samples on the three datasets; fixed proportion sampling is used for each category. (**a-1**) The OA of SA dataset. (**a-2**) The AA of SA dataset. (**a-3**) The Kappa of SA dataset. (**b-1**) The OA of UP dataset. (**b-2**) The AA of UP dataset. (**b-3**) The Kappa of UP dataset. (**c-1**) The OA of IP dataset. (**c-2**) The AA of IP dataset. (**c-3**) The Kappa of IP dataset. (**d-1**) The OA of LK dataset. (**d-2**) The AA of LK dataset. (**d-3**) The Kappa of LK dataset.

**Figure 15 sensors-24-03153-f015:**
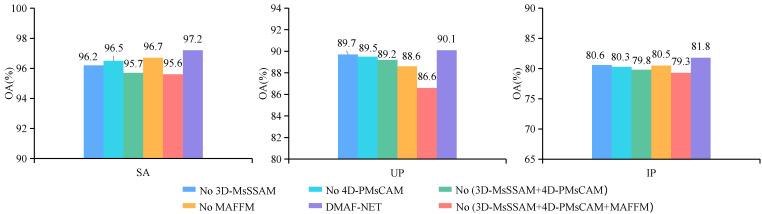
Classification results (%) of ablation experiments.

**Figure 16 sensors-24-03153-f016:**
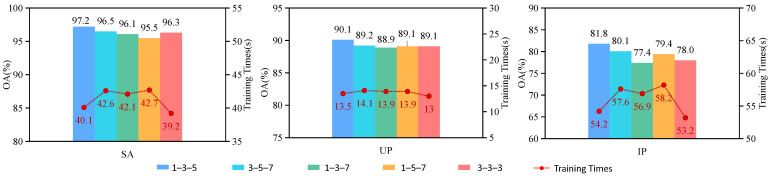
The influence of different size convolutional kernels in three branches of baseline.

**Figure 17 sensors-24-03153-f017:**
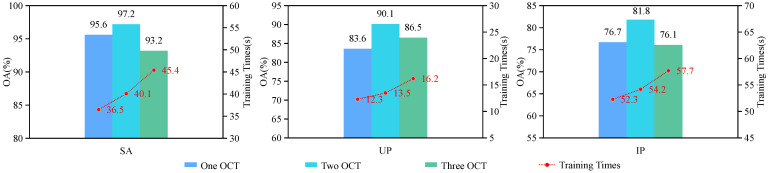
The influence of varying numbers of 3D octave convolutions in three branches of baseline.

**Figure 18 sensors-24-03153-f018:**
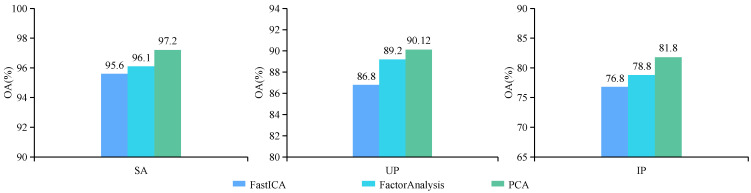
Classification results (%) of different dimensionality reduction method.

**Table 1 sensors-24-03153-t001:** Datasets employed during trials.

	SA	UP	IP	LK
Sensor	AVIRIS	ROSIS	AVIRIS	Headwall Nano-Hyperspec
Wavelength (nm)	400–2500	430–860	400–2500	400–1000
Spatial Size (pixels)	512 × 217	610 × 340	145 × 145	550 × 400
Spectral Bands	204	103	200	270
No. of Classes	16	9	16	9
Labeled Samples	54,129	42,776	10,249	204,542
Spatial Resolution (m)	3.7	1.3	20	0.463
Areas	California	Pavia	Indiana	Longkou

**Table 2 sensors-24-03153-t002:** Sample labels and sample sizes for the SA dataset.

No	MapColor	Class Name	Train Samples	Total Samples
5	10	0.1%	0.5%
1	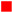	Broccoli_weeds1	5	10	2	10	2009
2	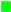	Broccoli_weeds2	5	10	4	19	3726
3	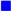	Fallow	5	10	2	10	1976
4	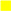	Fallow_rough_plow	5	10	1	7	1394
5	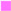	Fallow_smooth	5	10	3	13	2678
6	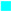	Stubble	5	10	4	20	3959
7	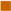	Celery	5	10	4	18	3579
8	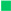	Grapes_untrained	5	10	11	56	11,271
9	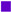	Soil_vineyard_develop	5	10	6	31	6203
10	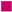	Corn_weeds	5	10	3	16	3278
11	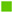	Lettuce_romaine_4wk	5	10	1	5	1068
12	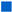	Lettuce_romaine_5wk	5	10	2	10	1927
13	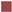	Lettuce_romaine_6wk	5	10	1	5	916
14	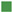	Lettuce_romaine_7wk	5	10	1	5	1070
15	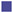	Vineyard_untrained	5	10	7	36	7268
16	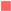	Vineyard_trellis	5	10	2	9	1807
		Total Samples	80	160	54	270	54,129

**Table 3 sensors-24-03153-t003:** Sample labels and sample sizes for the UP dataset.

No	MapColor	Class Name	Train Samples	Total Samples
5	10	0.1%	0.5%
1	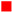	Asphalt	5	10	7	33	6631
2	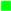	Meadows	5	10	19	93	18,649
3	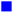	Gravel	5	10	2	10	2099
4	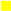	Trees	5	10	3	15	3064
5	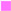	Painted metal sheets	5	10	1	7	1345
6	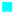	Bare Soil	5	10	5	25	5029
7	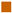	Bitumen	5	10	1	7	1330
8	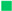	Self-Blocking Bricks	5	10	4	18	3682
9	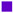	Shadows	5	10	1	5	947
		Total Samples	45	90	43	213	42,776

**Table 4 sensors-24-03153-t004:** Sample labels and sample sizes for the IP dataset.

No	MapColor	Class Name	Train Samples	Total Samples
5	10	5%	10%
1	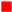	Alfalfa	5	10	2	5	46
2	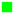	Cornnotill	5	10	71	143	1428
3	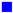	Corn-mintill	5	10	42	83	830
4	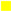	Corn	5	10	12	24	237
5	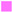	Grass-pasture	5	10	24	48	483
6	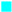	Grass–trees	5	10	37	73	730
7	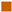	Grass-pasture-mowed	5	10	1	3	28
8	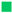	Hay-windrowed	5	10	24	48	478
9	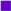	Oats	5	10	1	2	20
10	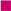	Soybean-notill	5	10	49	97	972
11	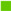	Soybean-mintill	5	10	123	246	2455
12	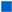	Soybean clean	5	10	30	59	593
13	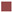	Wheat	5	10	10	21	205
14	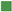	Woods	5	10	63	127	1265
15	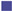	Buildings-Gra-Trees	5	10	19	39	386
16	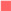	Stone-Steel-Towers	5	10	5	9	93
		Total Samples	80	160	513	1027	10,249

**Table 5 sensors-24-03153-t005:** Sample labels and sample sizes for the WHU-Hi-LongKou dataset.

No	MapColor	Class Name	Train Samples	Total Samples
5	10	0.1%	0.5%
1	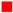	Corn	5	10	35	173	34,511
2	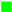	Cotton	5	10	8	42	8374
3	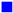	Sesame	5	10	3	15	3031
4	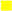	Broad-leaf soybean	5	10	63	316	63,212
5	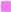	Narrow-leaf soybean	5	10	4	21	4151
6	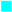	Rice	5	10	12	59	11,854
7	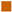	Water	5	10	67	335	67,056
8	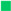	Roads and houses	5	10	7	36	7124
9	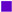	Mixed weed	5	10	5	26	5229
		Total Samples	45	90	204	1023	204,542

**Table 6 sensors-24-03153-t006:** The optimal spatial sizes and PCA components of the proposed model.

	SA	UP	IP	LK
Patch size	24 × 24	16 × 16	20 × 20	24 × 24
PCA components	24	24	44	16

**Table 7 sensors-24-03153-t007:** Classification results of various methods for the SA dataset with 10 training samples for each category.

Class No.	3D-CNN	HybridSN	SSRN	Tri-CNN	MCNN-CP	SSFTT	Oct-MCNN-HS	Proposed
1	**100.00**	99.33	96.46	99.60	**100.00**	99.97	99.93	99.80
2	99.06	98.67	99.98	98.57	99.00	99.89	**99.98**	99.78
3	99.44	99.33	94.29	99.38	99.50	99.91	99.90	**100.00**
4	99.30	**99.33**	79.90	98.30	98.00	98.82	98.33	98.99
5	92.76	96.03	98.58	97.59	92.67	95.94	91.82	**98.86**
6	98.89	96.33	**100.00**	98.30	99.50	98.86	99.13	99.13
7	99.19	99.67	99.85	98.23	**100.00**	99.91	99.67	99.34
8	81.71	74.67	90.66	80.87	79.83	85.94	79.70	**93.17**
9	99.74	**100.00**	94.66	99.60	99.00	99.87	99.98	99.46
10	94.21	95.67	89.39	95.87	96.50	**96.82**	96.42	95.95
11	97.22	**100.00**	98.13	98.02	**100.00**	99.89	99.87	**100.00**
12	99.03	94.67	**99.93**	96.50	95.83	96.28	98.77	98.96
13	99.45	98.67	**100.00**	97.67	95.17	97.81	98.86	99.37
14	98.77	98.87	98.03	98.90	97.83	**99.28**	97.78	97.30
15	71.73	81.67	65.93	80.74	79.83	84.66	89.35	**94.52**
16	97.07	99.13	96.16	97.19	98.67	**99.17**	97.54	97.70
OA (%)	91.20 ± 2.01	91.20 ± 1.07	91.10 ± 1.71	92.52 ± 1.97	91.90 ± 1.77	94.20 ± 1.07	93.40 ± 1.25	**97.20 ± 1.05**
AA (%)	95.50 ± 1.38	95.80 ± 0.90	93.90 ± 1.90	96.01 ± 1.95	95.70 ± 0.95	97.00 ± 0.84	96.70 ± 0.47	**98.30 ± 0.73**
Kappa × 100	90.25 ± 2.25	90.30 ± 1.19	90.00 ± 1.91	91.04 ± 1.90	91.00 ± 1.97	93.60 ± 1.19	92.70 ± 1.38	**96.90 ± 1.08**

The bolded value indicates the optimal value.

**Table 8 sensors-24-03153-t008:** Classification results of various methods for the UP dataset with 10 training samples for each category.

Class No.	3D-CNN	HybridSN	SSRN	Tri-CNN	MCNN-CP	SSFTT	Oct-MCNN-HS	Proposed
1	50.82	60.42	80.81	66.52	72.67	79.52	**83.02**	81.29
2	76.62	79.78	73.26	79.98	85.67	85.64	87.60	**93.22**
3	74.38	81.96	83.25	80.77	82.17	92.24	84.98	**90.04**
4	68.22	82.62	87.90	85.60	88.17	85.84	**91.22**	83.68
5	97.74	99.78	**100.00**	99.90	99.33	99.41	**100.00**	99.64
6	81.22	69.64	91.39	70.04	81.50	92.60	88.83	**95.40**
7	95.62	**99.40**	99.32	97.41	96.67	97.99	98.68	98.33
8	51.02	47.54	**94.78**	65.55	72.00	59.33	73.06	79.19
9	72.02	76.92	**99.93**	78.82	94.00	98.23	96.28	95.90
OA (%)	71.33 ± 3.04	74.20 ± 2.2	82.20 ± 0.99	82.20 ± 2.90	83.00 ± 1.67	84.67 ± 5.46	86.80 ± 1.47	**90.12 ± 1.09**
AA (%)	74.19 ± 3.53	77.60 ± 3.73	90.10 ± 2.20	82.90 ± 4.01	85.70 ± 1.56	87.87 ± 3.34	89.30 ± 1.33	**90.75 ± 1.23**
Kappa × 100	63.82 ± 3.89	66.90 ± 3.20	77.40 ± 1.34	75.97 ± 3.66	77.90 ± 2.01	80.30 ± 6.57	83.00 ± 1.90	**87.30 ± 1.19**

The bolded value indicates the optimal value.

**Table 9 sensors-24-03153-t009:** Classification results of various methods for the IP dataset with 10 training samples for each category.

Class No.	3D-CNN	HybridSN	SSRN	Tri-CNN	MCNN-CP	SSFTT	Oct-MCNN-HS	Proposed
1	99.40	97.62	**100.00**	98.52	**100.00**	99.53	98.61	98.15
2	37.34	42.02	52.19	60.12	66.00	64.46	**77.26**	67.98
3	54.31	58.62	58.23	59.02	65.60	**78.39**	78.37	77.03
4	79.23	84.44	79.88	84.84	96.40	95.72	**96.99**	93.83
5	76.99	81.56	85.90	83.59	88.40	85.52	84.50	**90.80**
6	92.18	95.28	86.78	95.98	96.80	97.32	**97.94**	94.37
7	99.80	99.82	**100.00**	99.98	**100.00**	**100.00**	**100.00**	**100.00**
8	96.47	99.62	86.86	99.60	95.80	93.27	96.44	**99.61**
9	99.80	99.94	**100.00**	99.99	**100.00**	**100.00**	**100.00**	**100.00**
10	64.55	67.02	75.29	68.12	77.80	85.68	**85.74**	80.27
11	52.03	53.42	53.05	60.42	66.80	64.36	58.47	**74.34**
12	39.89	56.62	42.65	56.65	58.60	69.94	**70.18**	70.14
13	99.49	96.62	99.23	94.69	97.80	99.82	**100.00**	96.92
14	81.63	76.82	93.55	79.77	88.00	94.62	85.77	**96.27**
15	56.41	85.42	81.74	85.40	81.40	87.45	**91.53**	90.78
16	99.00	95.20	**100.00**	97.21	94.20	99.60	**100.00**	97.19
OA (%)	62.77 ± 4.73	66.40 ± 3.16	68.70 ± 4.30	73.46 ± 3.76	76.50 ± 2.72	79.05 ± 2.95	78.70 ± 1.14	**81.80 ± 1.21**
AA (%)	76.81 ± 2.19	80.60 ± 1.93	80.10 ± 2.12	81.88 ± 1.93	85.90 ± 1.35	88.47 ± 1.67	88.90 ± 0.90	**89.20 ± 0.98**
Kappa × 100	58.72 ± 4.80	62.70 ± 3.42	64.70 ± 4.46	69.70 ± 2.49	73.60 ± 2.96	76.43 ± 3.18	76.10 ± 1.27	**79.50 ± 1.08**

The bolded value indicates the optimal value.

**Table 10 sensors-24-03153-t010:** Classification results of various methods for the LK dataset with 10 training samples for each category.

Class No.	3D-CNN	HybridSN	SSRN	Tri-CNN	MCNN-CP	SSFTT	Oct-MCNN-HS	Proposed
1	**99.99**	99.80	80.81	99.80	93.77	97.57	86.11	98.29
2	96.77	97.33	73.26	97.33	77.68	93.59	92.58	**99.49**
3	99.86	99.96	83.25	99.96	98.10	**100**	99.37	96.67
4	62.99	70.78	87.90	70.78	86.27	**93.12**	85.68	89.69
5	80.16	82.28	**100.00**	82.28	92.63	96.76	96.47	98.24
6	99.89	99.98	91.39	99.98	89.56	90.10	91.25	**100.00**
7	95.23	95.70	**99.32**	95.70	91.67	98.97	98.68	98.43
8	70.68	73.64	**94.78**	73.64	62.89	87.31	90.86	89.29
9	90.55	86.48	**99.93**	86.48	85.60	94.25	87.99	95.99
OA (%)	87.20 ± 3.00	87.38 ± 3.12	87.80 ± 2.99	91.88 ± 3.62	89.54 ± 2.45	94.99 ± 4.47	90.50 ± 3.17	**96.24 ± 2.09**
AA (%)	89.57 ± 3.66	89.40 ± 4.00	89.19 ± 2.18	92.41 ± 4.20	86.49 ± 2.36	94.47 ± 3.22	91.84 ± 3.03	**96.81 ± 2.13**
Kappa × 100	83.58 ± 2.86	83.03 ± 3.65	86.49 ± 3.01	90.05 ± 3.55	86.48 ± 3.01	93.48 ± 6.28	87.70 ± 4.01	**95.70 ± 1.89**

The bolded value indicates the optimal value.

**Table 11 sensors-24-03153-t011:** Total parameters and training times for all models on SA, UP, and IP datasets.

Model	SA		UP		IP		LK	
	Total Params	Training Time	Total Params	Training Time	Total Params	Training Time	Total Params	Training Time
3D-CNN	9,073,184	43.4 s	9,072,281	29.2 s	36,168,224	190.3 s	9,072,281	49.7 s
HybridSN	4,845,696	58.9 s	4,844,793	34.8 s	5,122,176	263.3 s	4,844,793	64.9 s
SSRN	749,996	1470 s	396,993	395 s	735,884	1440 s	760,155	1491 s
Tri-CNN	6,878,436	69.8 s	6,870,593	40.9 s	7,420,236	250.6 s	6,819,399	82.1 s
MCNN-CP	1,654,368	97.5 s	1,367,986	28.4 s	3,128,928	434.1 s	1,653,465	1249 s
SSFTT	153,224	5.9 s	152,769	5.8 s	153,224	5.3 s	153,621	6.5 s
Oct-MCNN	3,846,096	63.9 s	3,681,353	27.8 s	5,156,816	232 s	3,845,193	67.8 s
Proposed	2,932,878	40.1 s	2,604,295	13.5 s	2,778,254	54.2 s	2,717,895	46.1 s

## Data Availability

Data are contained within the article.

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
