# Peer review of "DMAF-NET: Deep Multi-Scale Attention Fusion Network for Hyperspectral Image Classification with Limited Samples"

_sensors, 2024, doi:10.3390/s24103153_

Round 1
Reviewer 1 Report
Comments and Suggestions for Authors
The article is more innovative and in line with the actual needs, and the scientific order of the experimental methods is reasonable, but there are some problems and suggestions:
Problems:
1.Corresponding Chapters: 2.3.2. 4D Pyramid-style multi-scale channel attention module
The overall structure diagram of the model presented in the chapter uses concatenate between the Attention Mechanism Unit and the Multi-attention feature fusion module, but adds when introducing the Multi-attention feature fusion module. If channel grouping is performed on the concatenate data, the details of the data should be described in the description of the Multi-attention feature fusion module. And the output result is add, but the output V diagram looks like the result of the concatenate operation, so it is recommended that the author change the color scheme.
2.Corresponding Chapters:3. Experiments and Results
In this section, the author's article mentions three datasets, but the actual one gives four datasets, and the abstract also mentions four datasets, so this is to be changed
3.Corresponding Chapters:3.4.2. Evaluation results with different training sample sizes
The article mentions four datasets, but the final evaluation results only show three datasets, why does the author only show the results of three datasets?
suggestions:
1.The corresponding section :2.3.1. 3D Multi-scale space-spectral attention enhancement module
The value of m is recommended for the author to give according to the actual model.
2.Corresponding chapters: 3.4.1. Evaluation results with a training sample limit of 10 for each category
It is recommended that the author bold the best values for each result in all result tables.
Reviewer 2 Report
Comments and Suggestions for Authors
In this paper, the authors propose a multi-scale attention fusion network to solve an important problem in the training of small-shot models in hyperspectral image classification tasks. In order to solve this problem, the neural network proposed by the authors includes a multi-scale feature extraction module, a pyramid multi-scale channel attention module, and a multi-attention fusion module. What’s more,a series of experiments were designed to prove the superiority of the proposed neural network from multiple perspectives. In general, the logic of this paper is rigorous and closed-loop, the experimental content is abundant with clear charts, which bring readers an excellent visualization experience. However, there still exist some problems to modify:
1. The manuscript requires further analysis of the reasons for the impact of different datasets on the results. In the selection of datasets, the manuscript selected multiple classical hyperspectral classification datasets(SA、IP、UP、LK). And when introducing them, the manuscript indicated that the purpose of selecting multiple datasets is to reflect the robustness of the model from multiple complex datasets. However, in subsequent experiments, the analysis of the impact of these different datasets was missing, and only the results on the datasets were listed. Experimenting on these datasets will make the manuscript’s content more fleshed out, but the results need to be analyzed in more detail, otherwise experiments with multiple datasets will be redundant.
2. Similarly, in 3.4.2 the manuscript mentioned different sampling strategies to study the impact of different training samples. How these two sampling strategies affect the experimental results needs to be further explained.
3. The reason for choosing the pyramid feature extraction method needs to be explained. This manuscript explains why most network models are chosen. In fact, there are still many other feature extraction methods, and what is the reason for the final choice of pyramid feature extraction.
4. Finally, in terms of format, colors in the table2 are unnecessary. Table 2 may be intended to correspond to the color of the plot in Figure 6, but such a color is not very pleasant to see.
Comments on the Quality of English LanguageThe English language is generally standard and regulated, but further modifications may be necessary.
Reviewer 3 Report
Comments and Suggestions for Authors
1.Introduction
In the presented scientific work, the introduction is performed at a high level. The evolutionary changes of convolutional networks are considered in detail, the problems arising when processing hyperspectral images are clearly identified. The main new directions in the development of algorithms are outlined, allowing for more accurate and efficient extraction and classification of features. The purpose of the work and the main results that were obtained during the implementation of this work are indicated.
Recommendations.
It may be worthwhile to more clearly structure the following points in the text, such as:
- The main purpose of this study is….
- The main tasks performed during this research are as follows...
2. Proposed Method
The section describes the general architecture and basic operating algorithms of the DMAF-NET network. The functions of individual modules in the network architecture are described in detail.
3. Experiments and results.
The authors provide detailed information about the datasets used. There is a detailed rationale for the choice of initial data sets on which the study was conducted, as well as criteria for evaluating experiments.
Comparative results obtained by other methods are presented, such as: 3D-CNN, HybridSN, SSRN, Tri-CNN, MCNN-CP, SSFTT, Oct-MCNN-HS. A justification is given for why comparisons were made with these methods.
4. Discussion
In this section, the authors evaluate the effectiveness of the proposed model, the rationale for the choice of individual blocks of the proposed network and their effectiveness based on the selected parameters. A comparison is made of the influence of various dimensionality reduction methods on the accuracy of the results obtained.
5.Conclusions
Brief conclusions on the work done are presented, the positive and negative aspects of the proposed method for processing hyperspectral images are indicated. Directions for further work to improve methods for processing hyperspectral images are proposed.
In this work, it is worth noting that a lot of detailed information is given on individual modules of the neural network, and the rationale for decisions made during research and when creating this method of processing hyperspectral images is presented. The effectiveness of individual modules and their impact on the overall accuracy and efficiency of the DMAF-NET network is analyzed. The weaknesses of the proposed method and further directions of research are shown.
The text of the work contains a small number of shortcomings, such as lack of punctuation marks and minor spelling errors.
Reviewer 4 Report
Comments and Suggestions for Authors
This manuscript provided interesting new model that can enhance accuracy of classification of hyperspectral image data.
There are several comments below.
Page 9, line 352: Section 3.1 needs to add how hyperspectral image data were prepared for running in the proposed model. And add more detail explanations about surface coverages of the four fields.
Page 10, line 362 and Table 1: Change IN to IP. Which one is correct, Indiana or Indian?
Page 10, line 373: In the caption of Figure 6, add LK.
In Figure 7 and 8, make same ranges of Y-axis.
Page 14, line 478: when the 7 models were run for evaluation, were same numbers of samples used? How many samples were? To calculate OA, AA, and Kappa for evaluation of models, which data was compared with, Pseudo color image or ground truth map?
From Figure 13 (3.4.2. Evaluation results with different training sample size), why filed LK was excluded? All 4 fields results should show up. If it is not, the reason should be added.
In Figure 13 and 14: if the scale of X-axis values are important in the model throughout the study fields, the charts should have a same range of X-axis.
Page 19, section 3.4.3: Total Parameters of the Proposed model are 4th or 3rd out of 7 models. Can you say that the proposed model reduced number of parameters efficiently?
In Figure 15: does DMSAN includes all 4 key components?
In Figure 15, 16, 17, and 18: make Y-axis same ranges.
Round 2
Reviewer 1 Report
Comments and Suggestions for Authors
The Authors modified the manuscript according to my comments and provided satisfactory replies. I think the manuscript can be considered for publication as it is.